# Mitigating Satellite-Induced Code Pseudorange Variations at GLONASS G3 Frequency Using Periodical Model

**Linyang Li** [1,2] , **Yang Shen** [2,*] **and Xin Li** [3]

1   School of Geodesy and Geomatics, Wuhan University, Wuhan 430079, China
2   Institute of Surveying and Mapping, Information Engineering University, Zhengzhou 450001, China
3   College of Computer and Information, Hohai University, Nanjing 211100, China
*   Correspondence: 631505020305@mails.cqjtu.edu.cn; Tel.: +86-1751-332-3992

**Abstract:** With the modernization of GLONASS, four M+ and two K satellites are able to broadcast code-division multiple-access signals at a G3 frequency. The evaluation of the G3 frequency is necessary, among which the satellite-induced code pseudorange variation is one of the most important indicators. Using the code-minus-carrier (CMC) combination, it was found that the magnitude of the code pseudorange variations at the G3 frequency is about 1 m, which is primarily caused by the fact that G3 is transmitted from a different antenna, the same as G1 and G2. However, different from BDS-2 medium Earth orbit and inclined geo-synchronous orbit satellites, the code pseudorange variations at the GLONASS G3 frequency have a very weak relationship with the elevation angle, while a strong correlation exists with the time series, by using wavelet transformation and correlation analysis. Validation is carried out using a single-site model and a continuous multi-site model over 24 h, and the correction performance of these two models is comparable. The systematic deviation of the CMC and Melbourne–Wübbena combinations are significantly corrected, so only random errors remain. With a more concentrated distribution of the pseudorange residuals of single point positioning, the standard deviation of the pseudorange residuals is reduced.

**Keywords:** GLONASS; code-minus-carrier combination; satellite-induced code pseudorange variations; time-dependent model; periodical correction model

## 1. Introduction

With the continuous development of satellite navigation applications, GPS, GLONASS, Galileo, the BeiDou Navigation Satellite System (BDS), and other satellite navigation systems have successively advanced modernization construction, and the new generation of navigation satellites is one of the most important aspects [1,2]. Since the original GLONASS-M satellites gradually reached the end of their service lives, Russia actively promoted the upgrading and modernization of GLONASS, and new satellites were supplemented. From 2001 to 2011, a total of four new GLONASS-M+ satellites were launched, and they were structurally optimized to achieve a longer design life. Then, in February 2011, the first K satellite was launched, and, in December 2014, the second K satellite was launched; these two GLONASS-K satellites adopted the latest pressure-free design. In addition to transmitting traditional frequency-division multiple-access (FDMA) signals, four GLONASS-M+ series satellites and two GLONASS-K satellites have also transmitted code-division multiple-access (CDMA) signals at the G3 frequency of 1202.025 Hz, since 2011 [3].

As shown in Table 1, as of November 2022, the GLONASS constellation consists of 24 satellites, including 18 GLONASS-M series satellites, 4 GLONASS-M+ series satellites, and 2 GLONASS-K series satellites, of which R26 is still undergoing testing. Many scholars have carried out signal-performance evaluations of the G3 frequency. Using the R21–R26 satellite pair, Zaminpardaz et al. [4] found a lower noise at the GLONASS G3 frequency than

that of GPS, and the double-difference ambiguity resolution and positioning performance were evaluated. Zaminpardaz et al. [5] also assessed the standalone GLONASS real-time kinematic (RTK) short-baseline positioning performance using both the FDMA and CDMA signals, and the positioning precision and ambiguity resolution success rate were improved by adding G3 signals. Zhang et al. [6] found that the signal-to-noise ratio (SNR) on the GLONASS G3 frequency is larger than that of the G1 and G2 frequencies. Additionally, they also found that GLONASS-M+ satellites have a significant inter-frequency clock bias (IFCB), and code pseudorange variations may exist at the G3 frequency. The code observables are affected by signal- and frequency-dependent delays, and delays caused by the nonsimultaneous transmission and/or reception of signals result in interfrequency and intersignal biases [7,8]; hence, the understanding and handling of the short-term variation of code biases is the prerequisite of high-accuracy positioning [9–11]. The satellite-induced code pseudorange variations are related to the satellite constellation configuration and satellite structure. In particular, the difference in the design of the antenna may lead to the reflection of the signal in the satellite body or influence internal hardware delay. All four GLONASS-M+ satellites and K satellite R26 transmit new L3 signals using a separate navigation antenna, while K satellite R09 is equipped with an improved phased-array navigation antenna covering all three frequencies [3].

**Table 1.** Constellation of GLONASS (as of November 2022). There are 18 M satellites, 4 M+ satellites, and 2 K satellites.

| Satellite Type | Satellite PRN |
| --- | --- |
| M | R01, R02, R03, R06, R07, R08, R10, R11, R13, R14, R15, R16, R17, R18, R19, R20, R23, R24 |
| M+ | R04, R05, R12, R21 |
| K | R09, R26 |

Previous research found that GPS SVN49 [12] and BDS-2 satellites present exceptionally large satellite-induced code pseudorange variations. For BDS-2 medium Earth orbit (MEO) and inclined geo-synchronous orbit (IGSO) satellites, Wanninger and Beer [13] established an elevation-dependent piecewise linear fitting method for an elevation range from 0° to 90°, and no receiver dependence and no station-location dependence were detected. Nadarajah et al. [14] studied the mixed-receiver BeiDou inter-satellite-type bias and its impact on RTK positioning. Using a low-frequency wavelet filter, Ma and Shen [15] determined the satellite-induced code pseudorange variation period of geostationary earth orbit (GEO), MEO, and IGSO satellites as 86,160 s, 86,158 s, and 46,391 s, respectively, which coincide with those of the corresponding satellite orbits. Additionally, using Fourier transform, correlation, and wavelet transform, the characteristics of the code pseudorange variations for BDS-2 GEO satellites over long periods were analyzed, and the precision of BDS-2 SPP was improved by correcting the observables with low-frequency variations on the previous day [16]. Ning et al. [17] further found that the code pseudorange variations of BDS-2 GEO C01, C02, and C04 had a great correlation with time series instead of elevation, while the C03 and C05 satellites were less affected by the code pseudorange variations. Beer et al. [18] explored the absolute GLONASS satellite antenna group delay variations (GDV) of triple observations, and inhomogeneous GDV curves were obtained due to a smaller number of satellites transmitting signals at G3. Additionally, due to changes in the structure of GLONASS-M+ and K satellites, it is necessary to further confirm whether satellite-induced code pseudorange variations exist, by conducting an analysis, modeling, and evaluation.

In this paper, our aim is to provide an insight into the GLONASS satellite-induced code pseudorange variations, wherein an analysis, modeling, and validation are the focus. This paper is divided into three main sections. Firstly, we introduce the methodology of the observable code-minus-carrier (CMC) combination. Secondly, the analysis and modeling

of the code pseudorange variations are presented. Finally, the effects are validated using the corrected model, and summaries and conclusions are provided.

## 2. Methodology

The code pseudorange variations are contained in the CMC combination and can be obtained using a combination of code measurement $P$ at frequency $i$ and carrier phase observables $\varphi$ at frequencies $i$ and $j$, which is also called multipath (MP) combination [19]:

$$\text{CMC}_i = P_i - \frac{f_i^2 + f_j^2}{f_i^2 - f_j^2}\lambda_i\varphi_i + \frac{2f_j^2}{f_i^2 - f_j^2}\lambda_j\varphi_j - B_i \tag{1}$$

where $\lambda$ is the wavelength of carrier phase with its frequency $f$. For convenience and following methods of Wanninger and Beer [16] and Li et al. [20], we set the frequency pairs obeying following principles: when case $i$ is G2 and G3, we set $j$ as G1; when $i$ is G1, we set $j$ as G2. The bias term $B_i$ is expressed as

$$B_i = \frac{f_i^2 + f_j^2}{f_i^2 - f_j^2}\lambda_i N_i + \frac{2f_j^2}{f_i^2 - f_j^2}\lambda_j N_j + D_c \tag{2}$$

where $N$ is the ambiguity, and $D_c$ represents the biases between the observables caused by hardware- and software-induced delays. Using the CMC combination, the satellite and receiver clock offsets, tropospheric delay, ionospheric delay, distance between the satellite and receiver, and other non-dispersive contributions can be eliminated. If no cycle slip occurs, $B_i$ is assumed to be a constant and is determined as average over raw CMC values for each continuous ambiguity block. Additionally, since the satellite antenna phase center difference between G1/2 and G3 frequencies reaches several decimeters, CMC variations have also absorbed this inconsistency. Therefore, CMC variations, which mainly reflect the code pseudorange variations and the inconsistency of the satellite antenna phase center, are obtained using Equation (1).

Theoretically, the maximum variation of carrier phase caused by multipath error is only one-quarter of the wavelength, which does not exceed 6.5 cm for GLONASS. Therefore, with respect to the decimeter-level or meter-level magnitude of code pseudorange variations, the variations of the carrier phase can be neglected. For this reason, long-term changes in the pseudorange observations can be found in the CMC time series, which are the code pseudorange variations of interest.

## 3. Experimental Results and Discussions

To analyze and model the GLONASS code pseudorange variations, GLONASS triple-frequency observations recorded from DOY 325 to 329, 2020, at 145 globally distributed stations (Figure 1) of the international GNSS service (IGS)'s Multi-GNSS Experiment (MGEX) network [21] were utilized. All these stations can receive the GLONASS triple frequency, and the data sampling rate is 30 s.

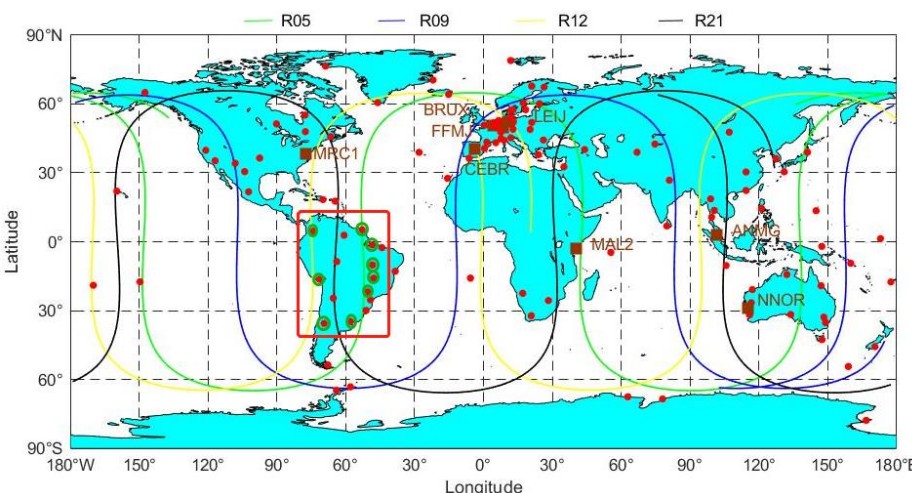

**Figure 1.** The distribution of 145 MGEX stations with GLONASS triple-frequency observations used in the experiment. The globally-distributed red points and brown squares represent stations that were used to model the code pseudorange variations and validate the established correction model. Ground tracks of R05, R09, R21, and R26 are given, and different color line denotes the ground track of different satellite.

### 3.1. Characteristic of the Code Pseudorange Variations

Since R04 rarely transmits the G3 signal, the code pseudorange variations of the remaining 3 M+ and 2 K satellites were studied. Moreover, since R26 is still currently undergoing tests, both the broadcast and precise ephemerides of R26 are missing; therefore, the elevation angle for R26 is not provided. Figures 2 and 3 show the relationship between the time series of the code pseudorange variations and elevation angle on DOY 325, 2020, at the BRUX and NNOR stations, respectively. Since the CMC combination only contains code pseudorange variations and observation noise, the observation noise fluctuates randomly around 0. If there is a systematic deviation in the CMC combination, code pseudorange variations occur. It can be seen from the figure that with the decrease in the elevation angle, the fluctuation range of the code pseudorange variations significantly increases. For these two GLONASS-K satellites, R09 and R26, no clear code pseudorange variations exist at the G1 and G2 frequencies. At the G3 frequency, the systematic deviation of the code pseudorange variations for R09 are very small, almost negligible, which is consistent with the study by Zhang et al. [6]. However, another K satellite, R26, differs from R09 at the G3 frequency, the code pseudorange variations of R26 vary with the elevation angle at NNOR station, and a constant upward tendency is observed at the BRUX station. The possible cause of this is that the first GLONASS-K satellite, R26, is equipped with a distinct G3 antenna, while the second GLONASS-K satellite, R09, uses an improved G1/G2/G3 phased-array navigation antenna [3].

For all the M+ satellites with a peak value of less than 1 m, code pseudorange variations are found at the G3 frequency, which is the same as the K satellites, and no obvious code pseudorange variations at the G1 and G2 frequencies are observed. Moreover, compared with the systematic "V-shape" trend of the code pseudorange variations of all the BDS-2 MEO satellites [22], the "V-shape" variation does not always apply to GLONASS MEO satellites; the code pseudorange variations of R05 are shown in Figures 2 and 3, of R26 are shown in Figure 2, demonstrating a constant upward or downward trend.

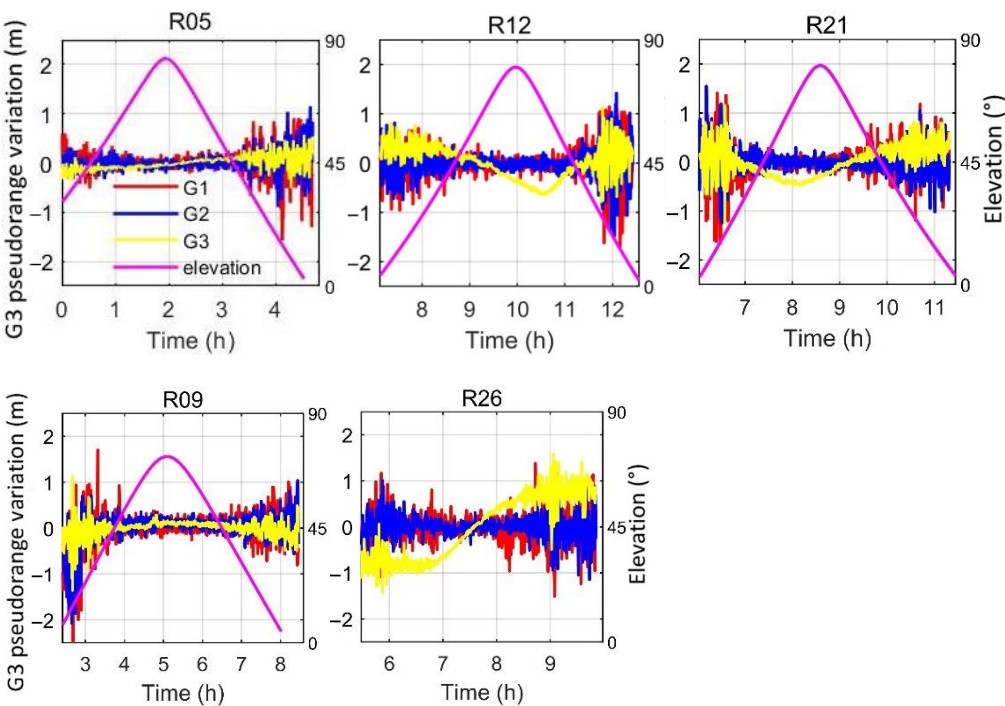

**Figure 2.** The relationship between the code pseudorange variations and elevation angle on DOY 325, 2020, at BRUX station. The red, blue, and yellow lines represent the code pseudorange variations on the G1, G2, and G3 frequencies, respectively. The purple line represents the elevation angle.

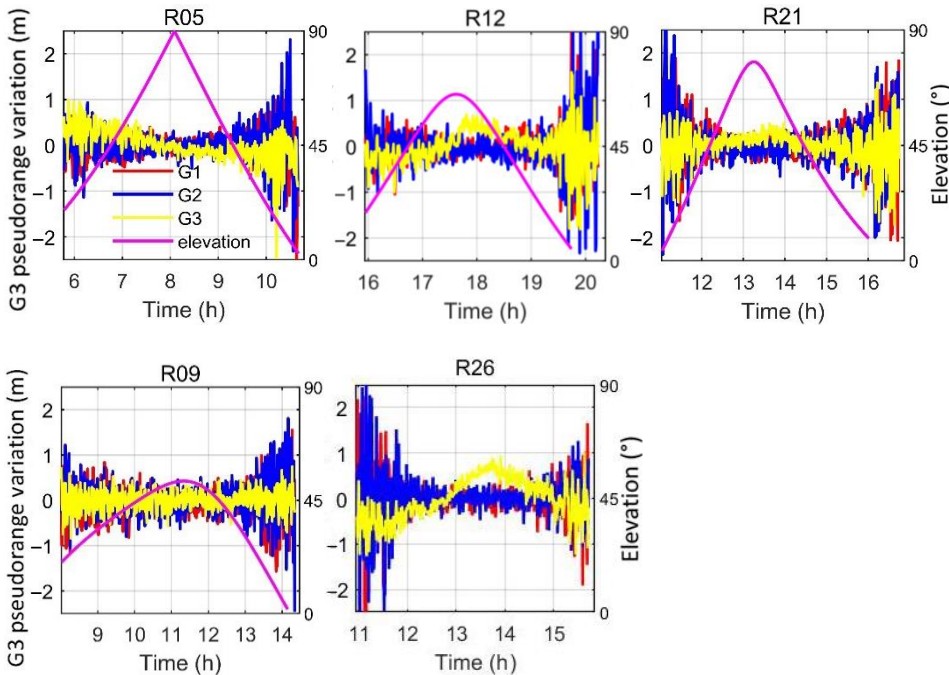

**Figure 3.** The relationship between the code pseudorange variations and elevation angle on DOY 325, 2020, at NNOR station. The red, blue, and yellow lines represent the code pseudorange variations on the G1, G2, and G3 frequencies, respectively. The purple line represents the elevation angle.

Moreover, for the R12 and R21 satellites, it is worth noting that there is an opposite relationship between the code pseudorange variations and the elevation angle estimated by the NNOR station and BRUX station. The BRUX station, which is located in the Southern Hemisphere, shows a negative correlation, while the NNOR station, which is

located in the Northern Hemisphere, shows a positive correlation. However, the correlation between the satellite-induced code pseudorange variations and the elevation angle of the BDS-2 MEO and IGSO satellites is always consistent and is unrelated to station distribution. This consistency makes modeling the code pseudorange variations of the BDS-2 MEO satellites relatively simple, but it will undoubtedly increase the difficulty of modeling the code pseudorange variations of the GLONASS-M+ and K satellites. However, a similar phenomenon has been observed in the BDS-2 GEO satellites, with a correlation between the code pseudorange variations and elevation angle that is also opposing, and a diurnal periodicity is found [17].

To further analyze the magnitude of the code pseudorange variations at the G3 frequency, the observation data from 145 MGEX stations on DOY 325, 2020, as shown in Figure 1, were utilized; the distribution of code pseudorange variations for all the M+ and K satellites is shown in Figure 4, wherein more than 400,000 code pseudorange errors were counted. It can be seen that the distribution of the code pseudorange variations is symmetric. The statistical results show that 99% of the code pseudorange variations are within ±2 m and more than 95% are within ±1 m. In addition, more than 12% of the pseudorange variations are around 0.

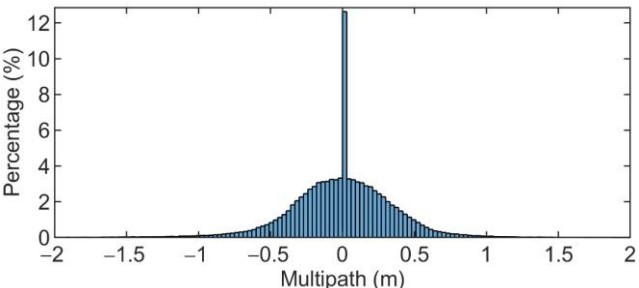

**Figure 4.** The distribution of the code pseudorange variations at the G3 frequency of all the M+ and K satellites. The observation data of 145 MGEX stations on DOY 325, 2020, as shown in Figure 1, were utilized, and more than 400,000 code pseudorange errors were counted.

### 3.2. Correlation Analysis of the Code Pseudorange Variations

For the investigation of the correlation of the code pseudorange variations, the Pearson correlation coefficient is utilized. The Pearson correlation coefficient is a statistic used to assess the monotonic correlation between two independent variables. Since a linear relationship between the variables is not required, the Pearson correlation is especially suitable when the relationship between the code pseudorange variations and the elevation angle or the time series is uncertain. In addition, Pearson correlation analysis is insensitive to extreme data, so the gross errors in the code pseudorange variations can be ignored [17]. Usually, the correlation degree of variables is judged according to the Pearson correlation coefficient. If the coefficient falls between 0 and 0.2, there is a very weak correlation or no correlation; 0.2–0.4 is a weak correlation, 0.4–0.6 is a moderate correlation, 0.6–0.8 is a strong correlation, and 0.8–1.0 is a very strong correlation.

### 3.2.1. Elevation-Dependent Modeling

Figure 5 shows the correlation coefficient between the code pseudorange variations and elevation angle at BRUX station on DOY 325, 2020. Except for GLONASS R21 and R09, the BDS-2 MEO satellite C14 was selected as a contrast. Figure 5 shows that, at the G1 and G2 frequencies of R21 and R09, the correlation coefficient between the code pseudorange variations and the elevation angle is almost 0. At the G3 frequency, the correlation coefficients of R21 and R09 are less than 0.4, representing a weak correlation. Additionally, compared with a correlation coefficient of larger than 0.8 for C14 at the B1 and B2 frequencies, which has been confirmed to be elevation-dependent, a coefficient of

0.4 is not enough to prove the existence of the correlation between the code pseudorange variations and the elevation angle of a GLONASS satellite.

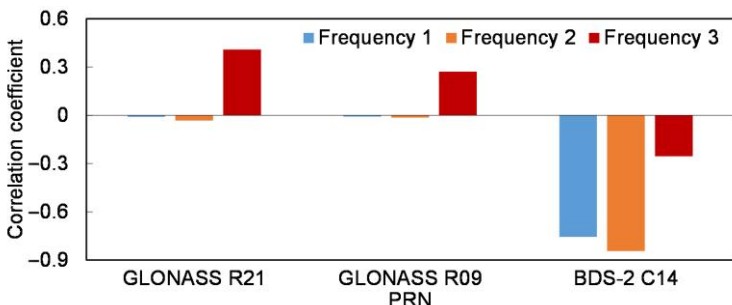

**Figure 5.** The correlation coefficient between the code pseudorange variations and elevation angle at BRUX station on DOY 325, 2020. The blue, orange, and red bars represent the correlation coefficients in the first, second, and third frequencies, respectively.

Furthermore, the observation data of all 145 MGEX stations, as shown in Figure 1, were utilized; Figure 6 shows the relationship between the code pseudorange variations and the elevation angle of R21. Figure 6 also shows that the code pseudorange variations are evenly distributed near the 0 value for each elevation angle, and no systematic deviation from 0 is found. Moreover, the Pearson correlation coefficient is only 0.0758, and, thus, the code pseudorange variations of R21 cannot be corrected by establishing the elevation-dependent model; this conclusion is the same for the other GLONASS-M+ and K satellites.

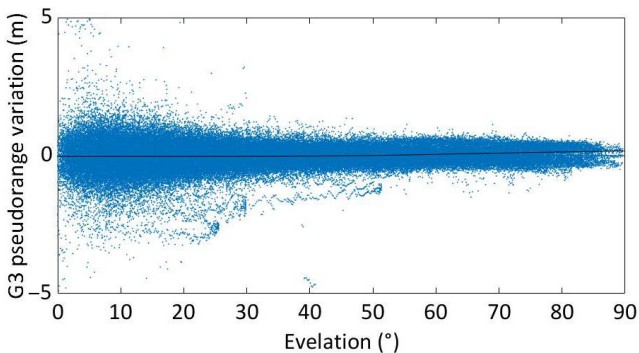

**Figure 6.** Relationship between the code pseudorange variations and the elevation angle of R21 on DOY 325, 2020.

### 3.2.2. Time-Dependent Modeling

The satellite-induced code pseudorange variations of BDS-2 MEO and IGSO satellites are both modeled and corrected using the elevation-dependent model, but this model is not suitable for GLONASS MEO satellites. Apart from the elevation-dependent model, the correlation between the code pseudorange variations and time series is analyzed, by referring to the periodical model used in the code pseudorange variations modeling of the BDS-2 GEO satellite [17]. Figure 7 shows the code pseudorange variations of R21 on DOY 325 and 326, 2020, at the BRUX station and FFMJ station, respectively. It can be seen that the code pseudorange variations for two consecutive days have a strong periodicity. The FFMJ station is close to the BRUX station, and the receiver and antenna at the FFMJ station are JAVAD TRE_3 DELTA and LEIAR25.R3, respectively, which are different from TRIMBLE NETR9 and TRM59800.00 at the BRUX station; however, the code pseudorange variations of R21 at the BRUX station and FFMJ station on DOY 325 and 326 essentially show the same trend. Therefore, as Wanninger et al. [13] pointed out, the code pseudorange variations are unrelated to the receiver and antenna type or the observation environment. Moreover, from

10:00 to 14:00 on DOY 325 and 8:20 to 12:20 on DOY 326, the code pseudorange variations decrease all the time, and a "V-shape" trend is not observed.

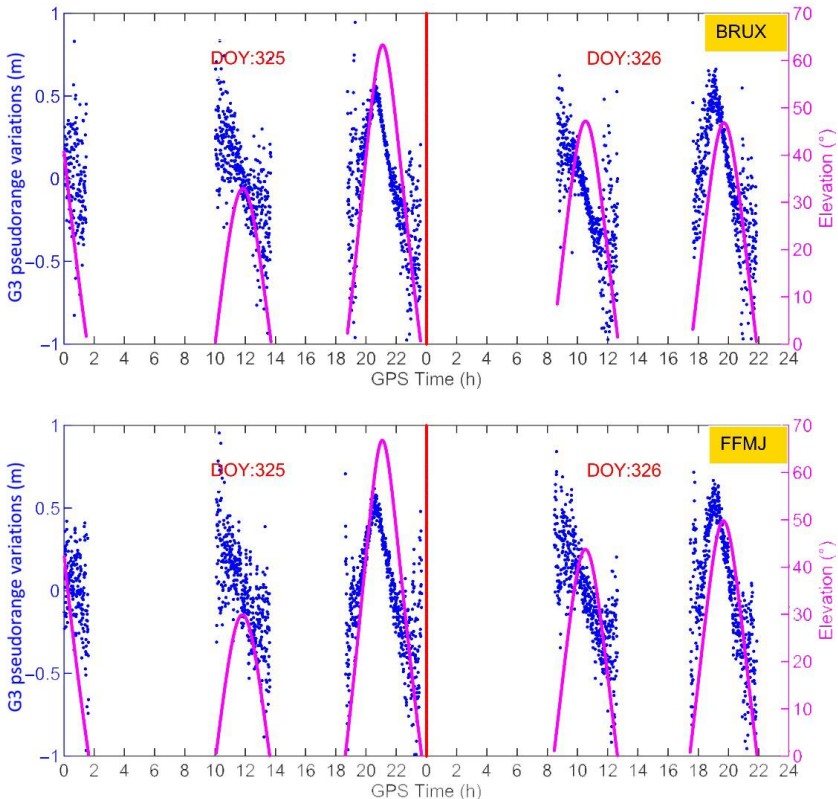

**Figure 7.** The code pseudorange variations of R21 on DOY 325 and 326, 2020, at BRUX station and FFMJ station. The blue dots denote the code pseudorange variations calculated by the CMC combination, the purple line denotes the elevation angle, and the vertical red line is the dividing line between DOY 325 and 326.

Figure 8 further shows the code pseudorange variations of R21 on DOY 325 and 326, 2020, at nine South American stations. (The nine red points with green circles in the red rectangle of Figure 1). It can be seen from the figure that the G3 code pseudorange variations of different stations are consistent and time-dependent.

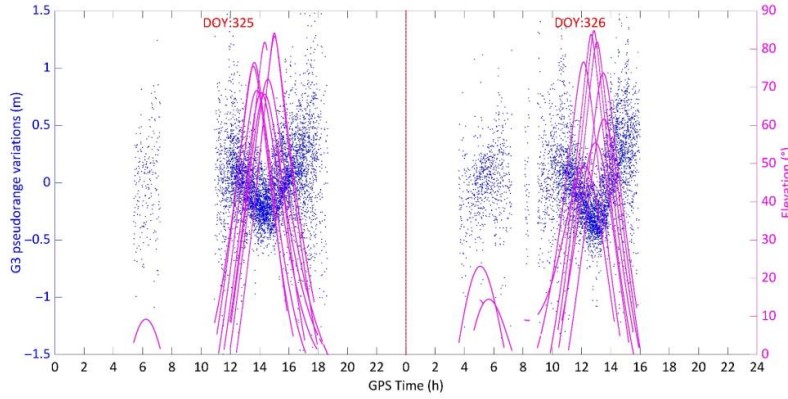

**Figure 8.** The code pseudorange variations of R21 on DOY 325 and 326, 2020, at nine South American stations. The blue dots denote the code pseudorange variations calculated by the CMC combination, the purple line denotes the elevation angle, and the vertical red line is the dividing line between DOY 325 and 326.

### 3.3. Modeling of the Code Pseudorange Variations Using Multi-Site

Since the MEO satellites cannot be tracked by one station all day, the 24 h time series of the satellite-induced code pseudorange variations cannot be determined using the observations of a single station. Therefore, to obtain the continuous code pseudorange variations of MEO satellites, all 145 stations shown in Figure 1 were utilized, and the code pseudorange variations at an epoch were determined using the elevation-dependent weighting method among multiple stations. There are both high-frequency components and low-frequency components in the code pseudorange variations, and the high-frequency components can be considered noise; therefore, Symlet wavelet transformation was utilized to extract the low-frequency components of the code pseudorange variations [16].

Figure 9 shows the code pseudorange variations of R21 on DOY 325, 2020, at 145 globally distributed stations. It can be seen that the G3 code pseudorange variations of different station are also not elevation-dependent.

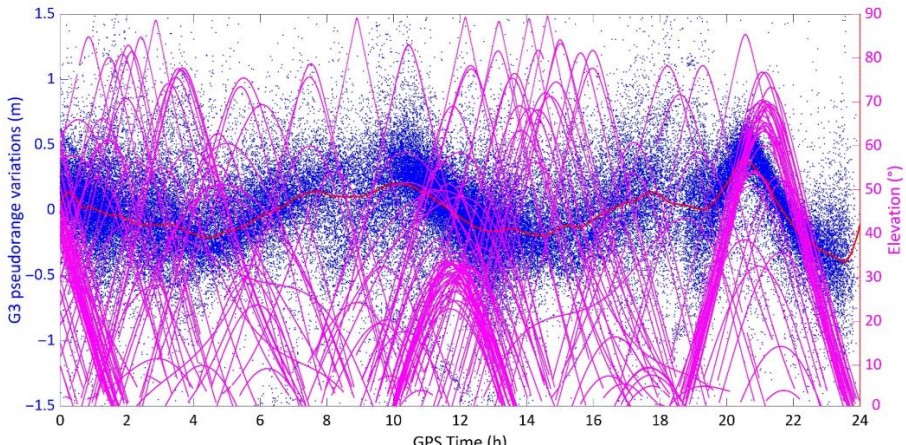

**Figure 9.** The code pseudorange variations of R21 on DOY 325, 2020, at 145 globally distributed stations. The blue dots denote the code pseudorange variations calculated by the CMC combination, the purple line denotes the elevation angle, and the red line represents the low-frequency components of the code pseudorange variations using wavelet transformation.

Moreover, Figure 10 shows the low-frequency components of the code pseudorange variations of R21 from DOY 325 to DOY 329, 2020. The figure shows that the variation trends for 5 consecutive days are periodic and consistent, and a striking day-to-day repeatability is observed.

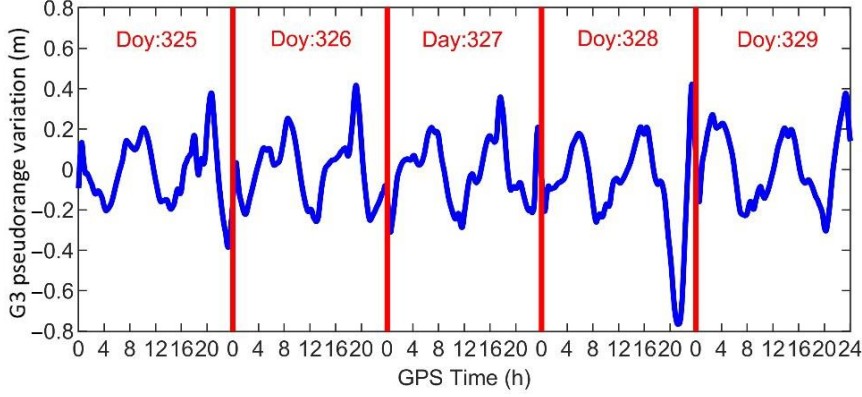

**Figure 10.** The low-frequency components of the code pseudorange variations of R21 from DOY 325 to DOY 329, 2020. The blue line represents the low-frequency components of the code pseudorange variations using wavelet transformation, and the red vertical line represents the dividing line between two consecutive days.

Then, a correlation analysis was carried out based on the low-frequency components of Figure 10, and the period was determined according to the peak value of the Pearson correlation coefficient. Figure 11 shows the Pearson correlation coefficient of the code pseudorange variations of R21 from DOY 326 to DOY 329, 2020. The statistical results show that the peak value of the Pearson correlation coefficient on DOY 326 is around 0.7 6, which is still larger than 0.5 over the following three days; therefore, a high correlation between the code pseudorange variations and time series was found.

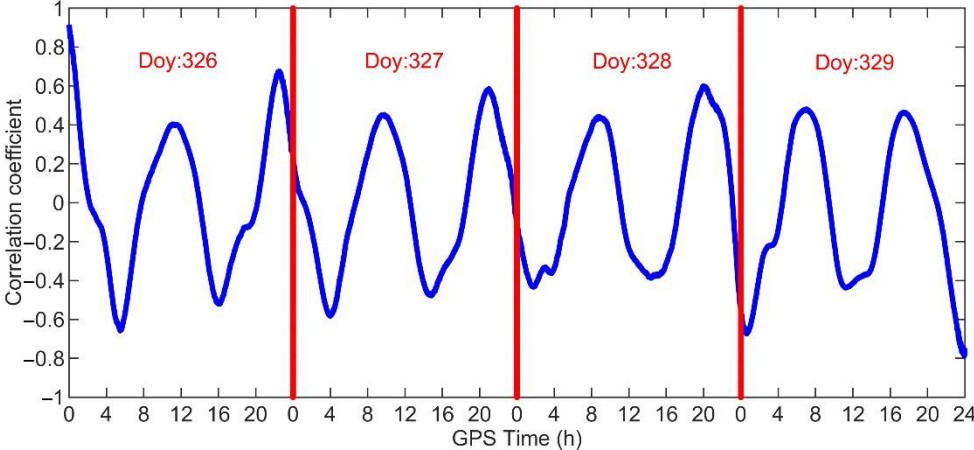

**Figure 11.** The Pearson correlation coefficient of the code pseudorange variations of R21 from DOY 326 to DOY 329, 2020. The blue line denotes the Pearson correlation coefficient, and the red vertical line is the dividing line between two consecutive days.

Moreover, the peak value of the Pearson correlation coefficient appears twice during one day, and the even-numbered peak is larger than the odd-numbered peak. Referring to Zhao et al. [22], a possible reason for this is that the code pseudorange variations of GLONASS MEO satellites might vary according to their relative geometries with the Sun. Since the orbit period of a GLONASS satellite is about 40,140 s, two periods are relatively close to the length of a sidereal day. Therefore, we selected an even-numbered peak as the period of the code pseudorange variations, with an average period of 80,520 s.

### 3.4. Validation of the Code Pseudorange Variations Model

Using the extracted low-frequency components of the previous cycle of the code pseudorange variations, the code pseudorange variations of the current period can be corrected, and the code pseudorange variations can be significantly decreased. The validation of the code pseudorange variations model was carried out using both the single-site periodical model and multi-site periodical model, wherein the CMC combination, Melbourne–Wübbena (MW) combination, and residuals of SPP were taken into consideration.

### 3.4.1. Correction Effect Using a Single-Site Periodical Model

Since the code pseudorange variations for two consecutive days show strong periodicity at a single station, validation was first carried out using the single-site periodical model. Observation data on DOY 325 and 326, 2020, at the LEIJ, FFMJ, BRUX, CEBR, ANMG, NNOR, MAL2, and MRC1 stations (the eight brown squares in Figure 1), were selected; Figure 12 shows the correction effects of R21. Figure 12 shows that, after removing the low-frequency components from the original code pseudorange variations, the systematic deviation of CMC combination caused by the code pseudorange variations is significantly corrected, and the distribution of the residual is random.

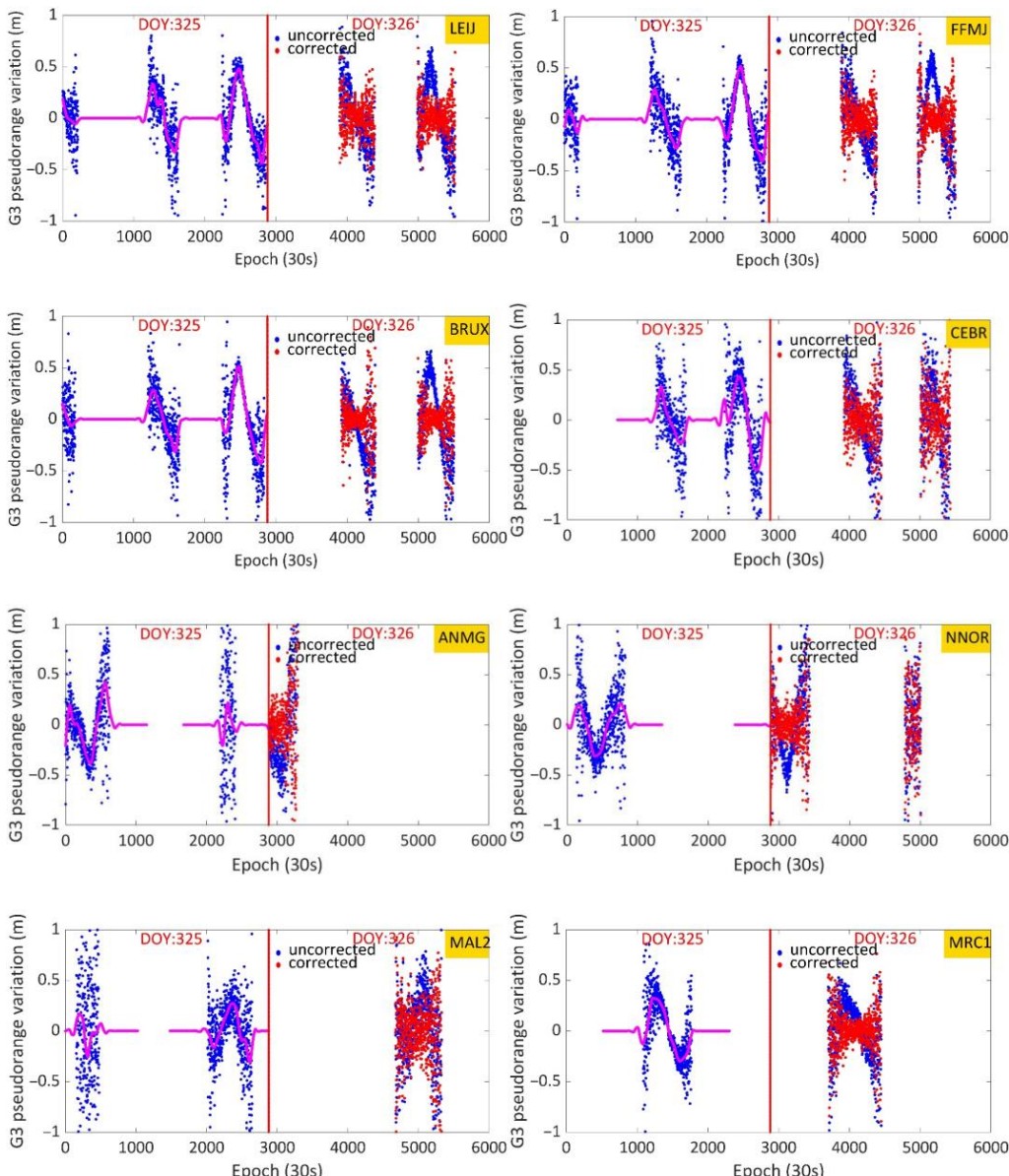

**Figure 12.** The code pseudorange variations of R21 on DOY 325 and 326, 2020, at LEIJ, FFMJ, BRUX, CEBR, ANMG, NNOR, MAL2, and MRC1 stations. The blue dots denote the original code pseudorange variations, the purple solid line denotes the low-frequency components of the code pseudorange variations using wavelet transformation, the red vertical line denotes the dividing line between DOY 325 and 326, and the red dots denote the corrected CMC combination.

Table 2 further gives the standard deviations (STDs) of the code pseudorange variations without and with correction at the aforementioned eight globally distributed stations. The statistical result shows that after using the single-site periodical model, the average STDs of the code pseudorange variations decrease from more than 21 cm to less than 11 cm, with a reduction of more than 44%.

**Table 2.** The STDs of the code pseudorange variations without and with correction at LEIJ, FFMJ, BRUX, CEBR, ANMG, NNOR, MAL2, and MRC1 stations.

| Station | DOY 325 | | | DOY 326 | | |
|---|---|---|---|---|---|---|
| | Without Correction (cm) | With Correction (cm) | Reduction (%) | Without Correction (cm) | With Correction (cm) | Reduction (%) |
| LEIJ | 20.90 | 12.02 | 42.49 | 19.72 | 11.08 | 43.81 |
| FFMJ | 20.13 | 11.26 | 44.06 | 20.16 | 10.88 | 45.95 |
| BRUX | 21.45 | 11.14 | 48.07 | 21.14 | 11.04 | 47.78 |
| CEBR | 20.68 | 12.42 | 39.94 | 20.88 | 12.24 | 41.38 |
| ANMG | 20.28 | 11.68 | 42.41 | 20.46 | 11.99 | 41.40 |
| MAL2 | 19.88 | 10.98 | 44.77 | 20.14 | 11.36 | 43.59 |
| MRC1 | 24.28 | 11.44 | 52.88 | 23.92 | 11.89 | 50.29 |
| NNOR | 22.28 | 12.48 | 43.99 | 22.87 | 12.95 | 43.38 |

3.4.2. Correction Effect Using a Multi-Site Periodical Model over 24 h

Although excellent correction effects have been achieved using the single-site model, it relies on the observation data from this station for the previous day or observation data from a nearby station [22]. A more general correction model is to apply the continuous low-frequency components determined by multi-site observations. We named this model a multi-site periodical correction model, as shown in Figure 11. Since this model is continuous over 24 h, it can be applied to the code pseudorange variations of worldwide receivers.

To verify the multi-site periodical model, Figure 13 shows the corrected CMC combination using the single-site periodical model and multi-site model on DOY 326, 2020, at the aforementioned eight globally distributed stations. It can be seen from Figure 13 that, after removing the low-frequency components of the code pseudorange variations determined by the 145 stations on DOY 325, the systematic deviation of the CMC combination on DOY 326 is corrected using the multi-site model over 24 h. Additionally, compared with the single-site model, a comparable performance is achieved using the multi-site model.

To further validate the correction effect of the multi-site periodical model, the MW combination, which was proposed by Hatch [23], was utilized. Figure 14 shows the time series of the uncorrected and corrected MW combinations at the aforementioned eight globally distributed stations on DOY 326, 2020. It can be seen from Figure 14 that the original MW function values reveal strong systematic variations in time, with amplitudes of more than half of a widelane cycle, which prevent precise ambiguity estimation and successful ambiguity fixing, and this is consistent with the BDS-2 MEO satellites [13]. After the application of the code corrections, these biases significantly decreased.

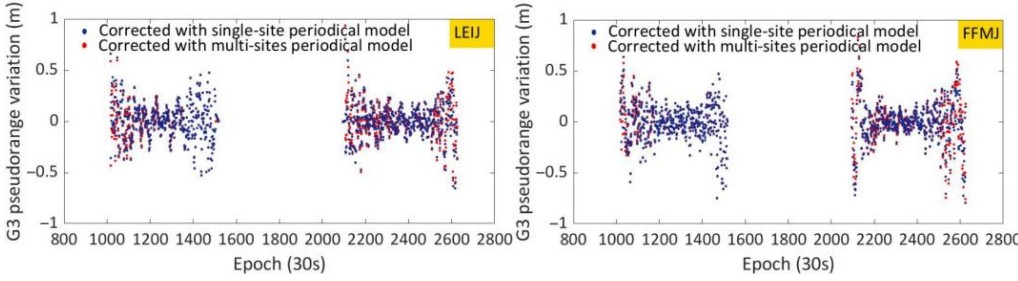

**Figure 13.** *Cont.*

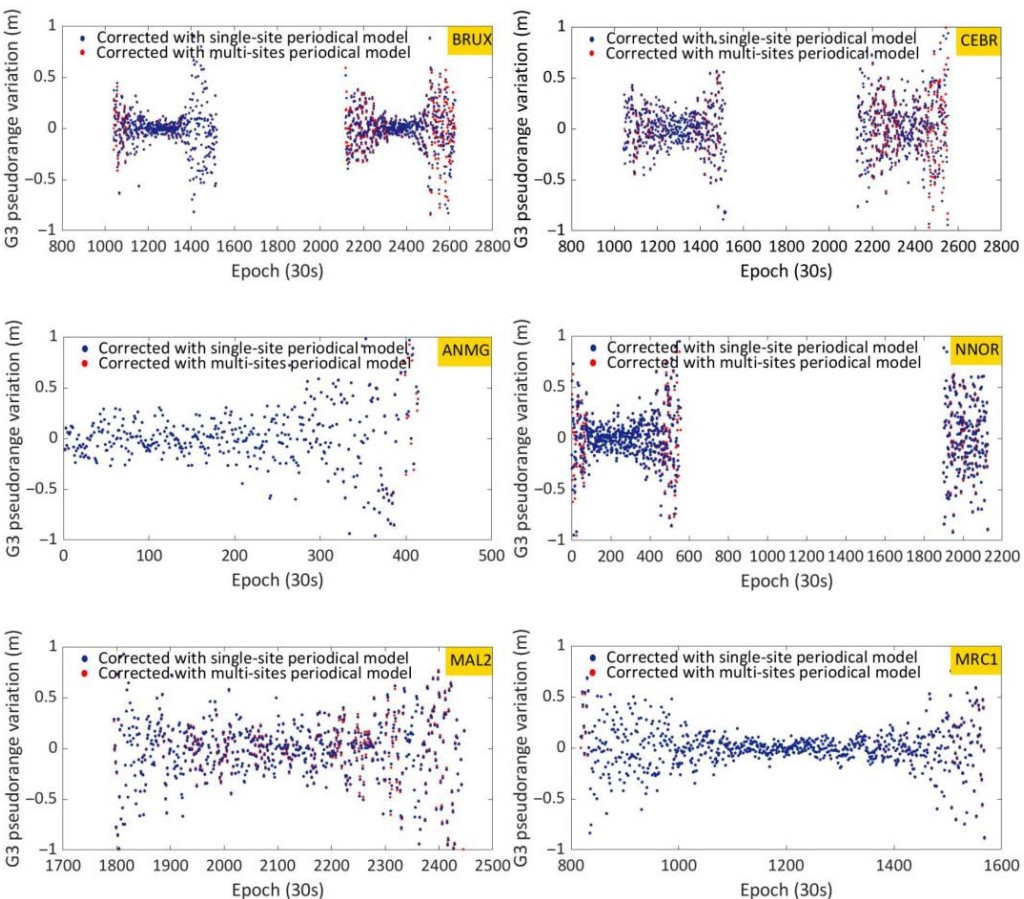

**Figure 13.** The corrected CMC combination of R21 on DOY 326, 2020, at LEIJ, FFMJ, BRUX CEBR, ANMG, NNOR, MAL2, and MRC1 stations. The red dots and deep blue dots denote the code pseudorange variations using the single-site model and multi-site model over 24 h, respectively.

As used in the study by Wang et al. [16], the multi-site periodical model was further verified using SPP. Figure 15 shows the original and corrected pseudorange residuals of R21 on DOY 326, 2020, at the BRUX station. Since the G3 signals of R09 and R21 were available at the BRUX station, the code pseudorange variations of R09 and R21 were corrected using the multi-site model, and a maximum of seven GLONASS satellites was available at each epoch. The differential code bias (DCB) was estimated and corrected, as conducted in the study by Li et al. [24]. It can be seen that the distribution of the corrected pseudorange residuals of R21 is more concentrated. The statistical results show that after using the multi-site periodical model, as used to correct the raw pseudorange observation, the STD of the pseudorange residuals reduced from 0.76 m to 0.36 m. In addition, the average positioning deviation of SPP has decreased from (1.85, 1.65, and 4.56) m to (1.63, 1.55, and 4.24) m.

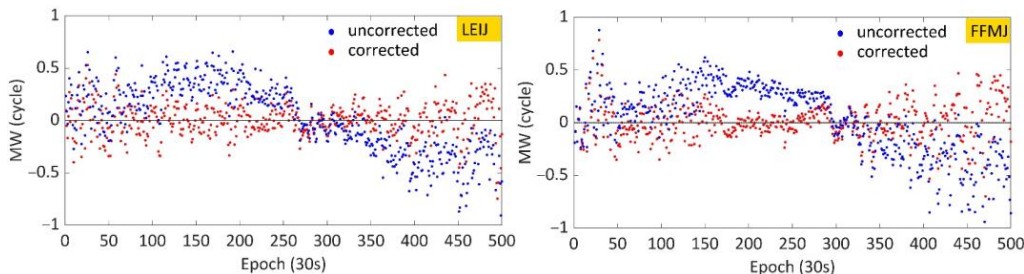

**Figure 14.** *Cont.*

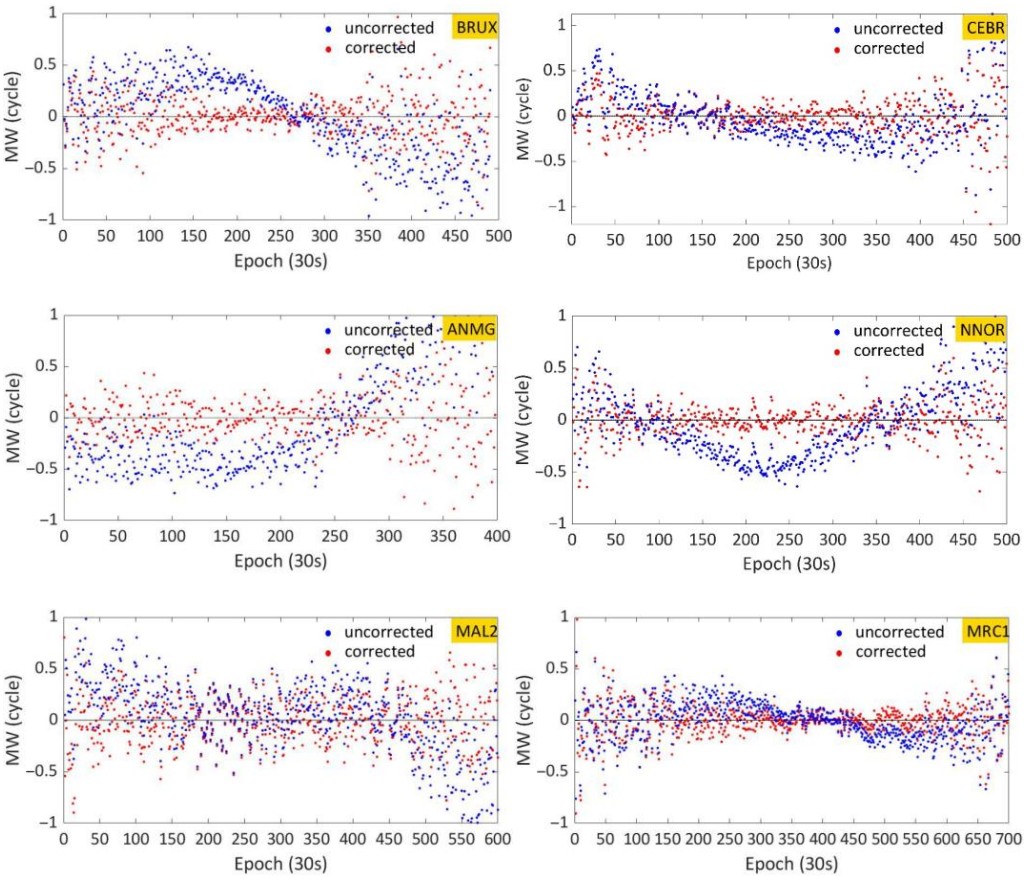

**Figure 14.** The MW combination time series on DOY 326, 2020, at LEIJ, FFMJ, BRUX, CEBR, ANMG, NNOR, MAL2, and MRC1 stations. The blue dots denote the original MW combination, and the red dots denote the corrected MW combination using the multi-site model.

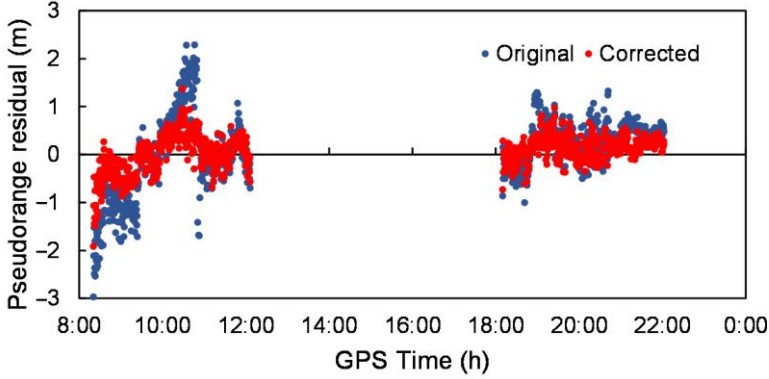

**Figure 15.** The pseudorange residuals of R21 on DOY 326, 2020, at BRUX station. The blue dots denote the original pseudorange residuals, and the red dots denote the corrected residuals using the multi-site model over 24 h.

## 4. Discussion

As part of GLONASS modernization, the Russian satellite system GLONASS has started sending signals using CDMA, and an evaluation of the satellite-induced code pseudorange variations at the new G3 frequency is being carried out. Since the antenna phase center offsets are not provided in the IGS official antenna file, the inconsistency of the position between the G3 antenna and G1/G2 antenna is not considered, i.e., the GLONASS-K satellite R26 still transmits CDMA signals on the G3 frequency next to the

FDMA signals on G1 and G2. As pointed out by Montenbruck et al. [3], the offset of the navigation antenna phase center between the G3 antenna and G1/G2 antenna can reach several decimeters, and, thus, the satellite-induced code pseudorange variations estimated in this paper absorb these phase center offsets. In contrast, by applying the offsets, Beer et al. [18] estimated the satellite group delay variations, which vary with the nadir angle. However, to correct the phase center offsets, the satellite body-fixed coordinate system should first be determined by considering the relative position with respect to the Sun and Earth, and an additional computational load will be imposed, which is not applicable for the massive, low-cost, low-power consumption chips for SPP. Hence, the proposed periodical correction model, which has absorbed the difference of the satellite antenna phase center offsets, is suitable for those massive SPP users.

## 5. Conclusions

This paper studies the satellite-induced code pseudorange variations at the GLONASS-G3 frequency, and it absorbs the inconsistency of the satellite antenna phase center between G3 and G1/2. The main conclusions are as follows:

1.  Compared with the systematic "V-shape" trend of the code pseudorange variations of all the BDS-2 MEO satellites, the "V-shape" variation does not always apply to the GLONASS MEO satellites; in addition, the correlation between the code pseudorange variations and the elevation angle for GLONASS satellites is both weak and opposing, which cannot be modeled using the elevation-dependent model applied to the BDS-2 MEO and IGSO satellites;
2.  The code pseudorange variations show strong periodicity, so a periodical correction model can be established; since a single-site periodical correction requires the observation data of the last day of this station or from a nearby station, the continuous multi-site periodical correction model over 24 h is more applicable;
3.  The validation of the code pseudorange variations model is carried out by using the single-site periodical model and multi-site periodical model; after removing the low-frequency components of the code pseudorange variations, the CMC combination, the MW combination, and the pseudorange residuals of SPP also cure this deficiency, so these two models can achieve comparable correction effects.

**Author Contributions:** L.L. and Y.S. provided the initial idea and wrote the manuscript; X.L. helped with performing the experiments; L.L., Y.S. and X.L. helped with analyzing the data. All authors helped with the writing, providing helpful suggestions, and reviewing the manuscript. All authors have read and agreed to the published version of the manuscript.

**Funding:** This study was supported by the National Natural Science Foundation of China (Grant Nos. 42104033) and the Postdoctoral Science Foundation of China (Grant Nos. 2022M712442).

**Data Availability Statement:** The GNSS data used in this paper were provided by the Multi-GNSS Experiment (MGEX) setup by IGS. The data from MGEX released by Institut Geographique National (IGN) can be accessed at ftp://igs.ign.fr/pub/igs/data/campaign/mgex/daily/rinex3 (accessed on 20 December 2021), and the data from MGEX released by Bundesamt für Kartographie und Geodäsie (BKG) can be accessed at ftp://igs.bkg.bund.de/IGS/obs (accessed on 20 December 2021).

**Acknowledgments:** The authors thank IGS MGEX for offering the GNSS data and products.

**Conflicts of Interest:** The authors declare no conflict of interest.

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
