# Peer review of "Mitigating Satellite-Induced Code Pseudorange Variations at GLONASS G3 Frequency Using Periodical Model"

_remotesensing, doi:10.3390/rs15020431_

Round 1
Reviewer 1 Report
The authors attempt to model the apparent GLONASS G3 pseudorange variation using a model that only depends on time. The literature on GLONASS G3 is still scarce, and I had great expectations about this manuscript. Unfortunately, it did not meet them. In my opinion, the paper cannot be accepted in its current form for two main reasons:
· The authors base their whole analysis on the CMC combination in equation (1). However, this equation is not applicable in the case where signals i and j are transmitted from different antennas in the satellite, which is the case for G3. The property that “distance between the satellite and receiver can be eliminated” (line 106-107) does not apply. The authors had at least to mention this explicitly and to explain the impact.
· As the apparent GLONASS G3 pseudorange variation is primarily caused by the fact that G3 is transmitted from a different antenna as G1 and G2, the effect depends on the satellite-receiver geometry. The conclusion claimed by the authors that this effect is the same for all receivers in the world at a given time (i.e. that it only depends on the time) is very questionable. The authors fail to provide evidence to support their claim, as they only provide data from two passes of a single satellite tracked by two selected IGS stations. The conclusion should be supported by a much stronger analysis.
There are also other flaws in the methodology, such as applying the Pearson correlation coefficient to check the dependence on the bias with elevation, as the Pearson correlation assesses “the monotonic correlation” (line 194), which is clearly not the case here.
Author Response
Dear Reviewer
Merry Christmas!
We gratefully thank you for taking the time to provide constructive comments and useful suggestions, which greatly improved the quality of the manuscript and enabled us to improve it. Every suggested revision and comment made by you has been accurately incorporated and considered. The following is a point-by-point response to your comments and the revisions are indicated. And the full text has been revised carefully once again.

Reviewer 2 Report
This paper aims to mitigate satellite-induced code pseudorange variations at G3 frequency of newly launched GLONASS satellites. Through using CMC combination, MW combination, and SPP, the established periodical correction model is verified.
There are some suggestions as following.
(1) From Figure 1, we can find that the trend of code variations are different for different GLONASS satellites. Since two types of satellites including M+ and K are involved, detailed reasons and classifications should be given to explain this phenomenon.
(2) In Figure 1, to show the relative position between the GLONASS satellites and ground stations more clearly, all GLONASS M+ and K satellites' ground tracks should be supplemented.
(3) The SPP performance and the code pseudorange variations of other GLONASS satellites with and without corrections are missing, these contents should also be given to show the correction effects of multi-site periodical model.
In the section "Discussion", if the PCO and PCV values at GLONASS G3 frequencies are provided by IGS official antenna file in the future, i.e. igs14.atx or igs20.atx, does the proposed periodical correction model need to be adjusted to accommodate the antenna phase center corrections?
Author Response

(The authors gave the same response as above.)

Round 2
Reviewer 1 Report
My main concern in the original manuscript was the applicability of a global bias correction model, that would only depend on the time, and not on the receiver location. In the new version, the authors added results from additional IGS stations. However, all those stations are located within the same region (South America for Fig. 8 or Europe for Fig. 12), which still does not address my concern.
In my opinion, it is essential to show better evidence of the applicability of the model at a global scale, as this is the main conclusion of the paper. This could be done by including distant stations in Fig 8 and Fig 12, i.e. show stations from different continents seeing the same satellite at the same time, but with a completely different satellite-receiver geometry.
Other remarks include:
* it would help to show the LEIJ, FFMJ and CEBR stations on the map in Fig 1, so that the reader can get an idea of the inter-station distance.
* Fig 4: please explain the peak at 0.
Author Response
Dear Reviewer
Once again, we gratefully thank you for taking the time to provide constructive comments and useful suggestions, which greatly improved the quality of the manuscript and enabled us to improve it. The three suggested revisions and comments made by you has been accurately incorporated and considered. The following is a point-by-point response to your comments and the revisions are indicated. And the full text has been revised carefully once again.
